# Addition of Vindoline to *p*-Benzoquinone: Regiochemistry, Stereochemistry and Symmetry Considerations

**DOI:** 10.3390/molecules26216395

**Published:** 2021-10-22

**Authors:** Shamsher Ali, Eric Hénon, Ritchy Leroy, Georges Massiot

**Affiliations:** 1H. E. J. Research Institute of Chemistry, International Center for Chemical and Biological Sciences, University of Karachi, Karachi 75270, Sindh, Pakistan; shamsher.ali@iccs.edu; 2Institut de Chimie Moléculaire de Reims, UMR CNRS 7312, Université Reims-Champagne-Ardenne, UFR Sciences, BP 1039, CEDEX 2, 51687 Reims, France; eric.henon@univ-reims.fr (E.H.); ritchy.leroy@univ-reims.fr (R.L.)

**Keywords:** vindoline, benzoquinone, atropoisomerism, cytotoxicity

## Abstract

Vindoline and catharanthine are the major alkaloids of *Catharanthus roseus* and are extracted in large quantities to prepare the pharmaceutically important Vinca type alkaloids vincaleukoblastine, vincristine and navelbine. The higher yield of vindoline relative to catharanthine makes it an attractive substrate for developing new chemistry and adding value to the plant. In this context, we have reacted vindoline with a selection of electrophiles among which benzoquinone. Conditions were developed to optimize the synthesis of a mono-adduct, of five bis-adducts, and of tri-adducts and tetra-adducts, several of these adducts being mixtures of conformational isomers. Copper(II) was added to the reactions to promote reoxidation of the intermediate hydroquinones and simplify the reaction products. The structures were solved by spectroscopic means and by symmetry considerations. Among the bis-isomers, the 2,3-diadduct consists of three unseparable species, two major ones with an axis of symmetry, thus giving a single set of signals and existing as two different species with indistinguishable NMR spectra. The third and minor isomer has no symmetry and therefore exhibits nonequivalence in the signals of the two vindoline moieties. These isomers are designated as *syn* (minor) and *anti* (major) and there exists a high energy barrier between them making their interconversion difficult. DFT calculations on simplified model compounds demonstrate that the *syn-anti* interconversion is not possible at room temperature on the NMR chemical shift time scale. These molecules are not rigid and calculations showed a back-and-forth conrotatory motion of the two vindolines. This “windshield wiper” effect is responsible for the observation of exchange correlations in the NOESY spectra. The same phenomenon is observed with the higher molecular weight adducts, which are also mixtures of rotational isomers. The same lack of rotations between *syn* and *anti* isomers is responsible for the formation of four tri-adducts and of seven tetra-adducts. On a biological standpoint, the mono adduct displayed anti-inflammatory properties at the 5 μM level while the di-adducts and tri-adducts showed moderate cytotoxicity against Au565, and HeLa cancer cell lines.

## 1. Introduction

Vindoline **1** is by far the most abundant alkaloid of *Catharanthus roseus* [1]. The molecule *per se* does not show any significant biological activity, but some combinations of vindoline with pharmacophores do present a suitable level of activity [2,3]. Vindoline is part of the so-called antileukemic Vinca alkaloids and, as such, has received considerable attention in coupling reactions with cleavamine type alkaloids. This approach was first proposed by Professor Atta-ur-Rahman, who suggested the use of catharanthine as the partner of vindoline in 1967 [4,5]. A breakthrough was made with Potier’s discovery that the Polonovski reaction was an efficient means of coupling vindoline and catharanthine to afford anhydrovinblastine en route to the other Vinca alkaloids [6]. Vindoline and catharanthine are extracted on an industrial scale in a process aiming at preparing the pharmacologically important navelbine. VLB and VCR are simultaneously obtained. Since the amount of vindoline exceeds the amount of catharanthine, it is of interest to find new uses for vindoline.

Vindoline is present in several bisindoles and in a few other natural products such as vindolicine [7], vindogentianine [8] and bannucine [9], for example. The biosynthesis of these molecules is based on the high nucleophilic character of C-10, which we found reactive towards aldehydes, ketals, orthoesters and quinones *inter alia*. In this article, we wish to describe the reaction with benzoquinone, which, depending on stoichiometry, gives a variety of adducts.

## 2. Results

The addition of nucleophiles to 1,4-benzoquinone is not a simple reaction. The initially formed products are dihydroquinones, which can be reoxidized by excess quinone to give substituted benzoquinones. Depending on time, reaction conditions and the stoichiometry of the reagents, the reaction may proceed further and yield a di-adduct, a tris-adduct and a tetrasubstituted benzoquinone (Figure 1). The addition reaction of amines and proteins with benzoquinone was studied long ago, with emphasis on the oxido-reduction properties of the adducts in biological media [10]. An excellent account of the possibilities offered by the addition of thio-nucleophiles to benzoquinones has been published by Katritzky et al. [11]. The reaction usually works under acid catalyst and recently triflic acid was proposed as a reagent of choice selectively giving mono adducts [12]. Depending on reaction time and stoichiometry, mono- or bis-adducts were obtained with Indium triflate in water [13].

### 2.1. Preparation of the Mono-Adduct ***3*** of Vindoline and Benzoquinone

In a first round of experiments, vindoline **1** was mixed with benzoquinone **2** in aqueous acetonitrile and nothing occurred (Table 1, entry 1). Addition of a base (NaOH) had no effect (entry 2). The reaction started to proceed with an acid catalyst, HCl (entry 3) or TFA (entry 4). After optimization of reaction conditions, mono adduct **3** was obtained in a 95% yield, when a 4 to 1 ratio of benzoquinone to vindoline and a large excess of TFA were used (entry 8, Figure 2). Compound **3** (see Appendix B) was a high melting point blueish solid, the MS of which corresponded to a 1:1 adduct ([M + H]^+^ at *m*/*z* 563.2393, calc. for C_31_H_35_N_2_O_8_
*m*/*z* = 563.2388). The UV showed the nearly unaffected three band pattern of vindoline (221, 250 and 309 nm) in overlap with the π-π* of benzoquinone, the n-π* band being shifted to 542 nm. The ^1^H NMR spectrum of **3** (see Appendix A) displayed all the signals of the vindoline moiety except those of the aromatic part where one of the aromatic protons was missing. The observation of two aromatic singlets at δ 7.04 and 6.34 ppm confirmed that the reaction expectedly took place at C-10 of vindoline. Three protons of the quinone formed the expected three spin system around 6.80 ppm, while the ^13^C NMR spectrum showed signals for the two carbonyls of the quinone at δ 186.5 and 188.0 ppm. All the other signals were assigned through the regular 2D NMR experiments (see Section 5).

### 2.2. Preparation of Higher Stoichiometry Adducts

A large part of the excess of benzoquinone required in the reaction is consumed in the reoxidation of the initially formed dihydrobenzoquinone, a reaction that must be faster than the initial addition since the intermediate does not accumulate. To facilitate this reoxidation and to control the consumption of benzoquinone, Cu(II) was added to the reaction mixture, as proposed by Bäckvall for the reoxidation of benzoquinone in Pd catalyzed reactions [14]. Table 1 and Figure 3 summarize these results. The use of a large excess of vindoline provided the high molecular weight tri-adduct **7** and tetra-adduct **8**, while the optimum ratio for the production of the di-adducts (**4**–**6**) was slightly over 2. The different adducts were easily distinguished by TLC and the separations were achieved by column chromatography. Structures were determined by spectroscopic means and symmetry considerations.

#### 2.2.1. The Bis Adducts. The Simple Adducts and General Considerations

As expected, three bis adducts, **4**, **5** and **6** were obtained with **5** being by far the most abundant. For the sake of clarity, they are represented below under the simplified drawings in Figure 1, where V stands for vindoline. They all showed the expected molecular ion at *m*/*z* 1017 [M + H]^+^ corresponding to a C_56_H_65_N_4_O_14_ formula, that is to say to the substitution of two hydrogen atoms of benzoquinone by vindolines.

Compound **4** is a centrosymmetric bis-adduct and as expected, it showed a single set of signals in the ^1^H and ^13^C NMR spectra. It is a pink solid with an extended UV chromophore (λ_max_ at 526 nm). The low field part of the ^1^H NMR spectrum showed four two proton singlets for H-3′, H-9, H-12 and H-17 at δ 6.76, 6.35, 7.04 and 5.38. The quinone proton was identified by observing long range coupling with the ketone carbonyl at δ 188.3; it also coupled to two quaternary carbons at 114.6 and 146.7 respectively assigned to C-10 (vindoline) and C-2′ (quinone). All the vindoline signals were identified by comparison with the spectra of **1** and by suitable 2D NMR experiments.

In compound **5** (see Appendix C), the two vindoline moieties are exchangeable through an axis of symmetry and therefore give a single set of signals. This is not the case for the quinone carbonyls, which showed two signals at δ 189.6 and 187.8. In the ^1^H NMR spectrum, the four singlets were at δ 7.02 (H-12), 6.75 (H-3′), 6.34 (H-9) and 5.37 (H-17). Due to the symmetry H-3′ showed couplings in the HMBC experiment with C-5′ (equivalent with C-3′), C-10 (115.5) and C-2′ (148.7). In the UV spectrum the n-π* band was shifted to 550 nm indicating extension of the chromophore compared to compound **4**.

The ^13^C NMR spectra of compounds **4** and **5** show a high degree of similarity but can be easily distinguished by their quinone carbonyl resonances. The ^1^H NMR spectra are not very different either, with noticeable and unexplained differences being observed for CH_3_-18 (0.05 ppm), H-14 (0.03 ppm) and H-21 (0.06 ppm).

#### 2.2.2. The Vicinal Bis Adduct **6**

Since the structures of compounds **4** and **5** are firmly established, there only remained the possibility of vicinal substitution for compound **6**. Despite our efforts, and even though UPLC showed a highly predominant peak, compound **6** could not be completely purified, as judged by NMR. The mass spectrum showed the expected molecular ion and the UV was shifted to 534 nm. The ^1^H NMR spectra showed a major compound with the four typical singlets of the bis-adducts at δ 6.89, 6.30, 6.28 and 5.35 for H-5′ (H-6′), H-9, H-12 and H-17 respectively. These resonances were assigned as above by means of HSQC and HMBC experiments. In addition, the spectrum showed several other sets of singlets of smaller intensities, which could be assigned to other isomers or conformers.

#### 2.2.3. Vicinal Bis Adducts and Symmetry Considerations

The di-substitution and the bulky nature of vindoline makes rotation around the C-10- C-2′ bond difficult and it might be that the vicinal di-adducts present a character of atropoisomerism. In this case, four vicinal di-adducts may be expected to form: two *syn* and two *anti* isomers. The *anti* adducts exist as two isomers while the *syn* adducts are one and the same compound (homomers). They will be denominated **6a1**, **6a2** and **6b**. Figure 2 is a simplified representation of these molecules in which, for the sake of clarity, an anisole ring is drawn instead of the vindoline. Compounds **6a1** and **6a2** possess an axis of symmetry passing through the middle of the C-2′-C-3′ and C-5′-C-6′ of the benzoquinone and exchanging the vindolines, which, therefore must present a unique set of signals. Due to the natural chirality of vindoline, compounds **6a1** and **6a2** are different. There is no element of symmetry in compound **6b** and the two vindolines must give different sets of signals of respective equal intensity.

#### 2.2.4. The Vicinal Adducts and NMR Assignments

Figure 3 is another simplified representation of the three vicinal dimers in which the vindolines are represented by the boxes marked V, where H-2 serves as a marker of the asymmetry of vindolines. This figure shows that the three structures are different, which is not obvious for **6a1** and **6a2**, where the H-2 atoms are either inside or outside the branches of the U shaped molecules.

As far as NMR is concerned, compounds **6a1** and **6a2** are expected to display a single set of resonances for the vindolines, with no reason for them to be identical, while **6b** should show signals for two sets. As an example, H-2 may show up to four distinct resonances and the same is true for other observable signals such as those of CH_3_-18 and H-15.

#### 2.2.5. Higher Order Adducts

It took forcing conditions to obtain the higher order adducts **7** and **8**, i.e., a large excess of vindoline and a longer reaction time. For each of these compounds, there is only one possible regioisomer.

The tri-adducts may be seen as arising from the addition of a vindoline unit to vicinal bis-adducts **6a1**, **6a2** and **6b**, even though the intermediacy of di-adduct **5** is the most probable event. Compounds **6a1**, **6a2** are different and they are expected to only give two tri-adducts: **7a1** and **7a2,** since the positions on the quinone are equivalent. In the *syn* adduct however, the two quinone positions are different and therefore, one may expect two compounds **7b** and **7c**. This is represented in Figure 4 with the same simplified drawings as in Figure 2. As regards symmetries, none of the four adducts possess any element of symmetry and therefore each of the three vindolines is expected to give different signals. It is not easy to disentangle the resonances in the ^1^H NMR spectra, but at high field and for the major compound a set of three triplets of equal intensities is observable (Figure 5).

The situation with the tetra-adduct is more complicated with seven possible isomers: a single compound **8a** with all vindolines oriented in the same manner, two compounds **8b** and **8c** with one vindoline oriented towards the other face with regards to the three others and four isomers with two vindolines oriented on the same side (2,3, 2,6, 2,5 and 3,6 with respect to the quinone) **8d**–**g**. Isomers with the OMe top-oriented 2,5 and 3,6, **8f** and **8g** are not superimposable and therefore are different compounds) (Figure 6 and Figure 7).

In isomer **8a** since vindoline is chiral, none of the four vindolines is directly exchangeable through a plane of symmetry. However, if the molecule is rotated by 180° along the yy’ axis and then by 180° along the xx’ axis, then **8a** is recovered unchanged (Scheme 5). V1 and V3, on one hand, V2 and V4, on the other, are thus exchanged and are equivalent (Figure 4). The NMR spectra of compound **8a** should show two sets of signals for vindoline but a single carbonyl for the benzoquinone.

There are two tetra-adducts **8b** and **8c** with only one vindoline moiety pointing downwards with the three other ones being upwards (Figure 6). Observation of **8b** and **8c** from the ketone side with the two vindolines with their OMe upward, brings the vindoline with Ome pointing downward either on the right or on the left side. They are not exchangeable since vindoline is chiral. Compounds **8b** and **8c** are therefore different and within each of them, the four vindolines are different and thus these isomers may be expected to show four sets of vindoline signals of equal intensity. A consequence of the absence of symmetry is that two signals should be detected for the benzoquinone carbonyls.

In isomer **8d**, a rotation along the yy’ axis exchanges V1 and V4, V2 and V3, which are therefore equivalent pairs. There is no mechanism of exchange for the pairs V1, V2 and V3, V4. This isomer would be expected to give two sets of signals and a single benzoquinone carbonyl. In isomer **8e**, the vindolines on the same side of the ketones are exchanged by a 180° rotation around axis xx’ and are therefore equivalent. There is no symmetry to exchange the other pairs and this molecule is expected to show two sets of signals for the four vindoline moieties. The quinone carbonyls are also exchanged in this rotation and should give a single signal. Finally in the last isomers **8f** and **8g** rotations around the two axis xx’ and yy’ exchange the four vindoline units and symmetry considerations suggest that these isomers will show a single set of signals and a single benzoquinone carbonyl (Figure 7).

#### 2.2.6. The NOESY Experiment: Further Complexity and DFT Calculations

The NOESY maps obtained for the simple compounds **3**, **4** and **5** showed the expected correlations of positive intensity for the vindoline protons interactions. The mixture of di-adducts **6**, however gave positive and negative NOEs. This was rather unexpected since these molecules of the same molecular weight should have close correlation times (τ_C_). The phenomenon was particularly intriguing in the high field area, which showed correlations between methyl groups, negative in sign (same as the diagonal) (Figure 8). In other parts of the spectra, regular (positive) NOEs were observed leading to conclude that these negative effects are due to a slow exchange process between conformers. The same situation happened with the trimer, but due to the relatively high molecular weight (1470), all correlations were negative. The existence of conformers stable enough to give rise to exchange at the NMR timescale, makes definitive assignments of signals to a particular species extremely difficult even for compound **6**, which exists as three forms with two conformations each, amounting to eight different sets of signals for vindolines. Of course, the situation is even worse for the tri- and tetra-adducts.

Density Functional Theory (DFT) M06-2X/6-311G** calculations were performed in methanol (implicit solvent) on simplified models corresponding to structures **6a1**, **6a2** and **6b** of Figure 2. In a first round of calculations, it was demonstrated that rotating one anisole ring by 180° around the single quinone-anisole bond involves a substantial energy barrier (ΔG°^‡^(298 K) = 21.6 kcal·mol^−1^). Actually, the *ortho* methoxy group knocks into the quinone carbonyl (steric repulsion), making the rotation difficult without a deformation of the quinone ring. Thus, the interconversion between **6a** and **6b** is certainly not possible under our observation conditions. Since an eventual interconversion between **6a1** and **6a2** must occur through the intermediacy of **6b**, **6a1** and **6a2** are definitively different molecules. At that stage the NOESY exchange correlations could not be explained and it was decided to perform a potential energy surface (PES) scan of the conrotatory motion in which both anisole groups turn simultaneously in the same direction. The results showed that there exist for each isomer (**6a1**, **6a2**, **6b**) two distinct minima on the PES, the oscillation between them resembling a windshield wipers motion. In the case of the *syn* isomer, these two conformations are chemically equivalent while they are not for the *anti* since in that case either the methoxy or the phenyl comes close to each of the quinone carbonyls. The transition between these conformations does not require much energy (ΔG°^‡^(298 K) in the range 1.3–2.5 kcal·mol^−1^) and thus is observable at the NMR time scale. 

## 3. Discussion

The addition of vindoline to benzoquinone is not different from that reported for amino and sulfur nucleophiles [10,11]. A mono adduct is formed first, followed by a di-adduct, either -2,4 or -2,5 in the case of sulfur and -2,5 in the case of nitrogen. If nothing is anticipated regarding the reoxidation of the initially formed dihydro-benzoquinones, the reaction mixtures are far more complicated and minor adducts may not be detected. It is worth noting that the 2,4 or -2,5 adducts are not easy to distinguish and ^13^C NMR, used here as a diagnostic tool, is not always reliable since, for example 2,5- and 2,6-bis(cyclohexylsulfanyl)(1,4)benzoquinones both showed a single carbonyl signal [11].

At variance with heteroatom nucleophiles, simple heterocycles seemed to give predominantly the 2,3-adducts and in a remarkable article by Escolastico et al. the presence of separable atropoisomers is described [15]. The difference in behavior with vindoline may be explained by the steric bulk in the mono-adduct, which prevents the approach of the second molecule of vindoline from the same side, even though the trajectory must be perpendicular to the plane of the benzoquinone. There were also differences in operating conditions: room temperature, acid and Cu(II) catalyst in our case, compared to no catalyst and reflux in dioxan, in the reference [15]. 

Vindoline does not follow the tendency of other nucleophiles and the 2,5-adduct is predominantly obtained as the second intermediate. Then, following what was observed for the bis adducts, the third addition occurs from the opposite face of the already present vindolines to give the *anti* compound **7a**. This compound may be considered the major starting material for the synthesis of the tetra-adduct and should lead to the predominant formation of isomers **8d** and **8e** in equal proportions.

In summary, the reaction of vindoline with benzoquinone could lead to the formation of seventeen compounds. Three of them (**3**, **4**, **5**) were separated and characterized in a pure state and two pairs (**6a**, **6b** and **7a**, **7b**) were separately characterized in the same NMR tube. The seven isomeric tetra-adducts could not be distinguished due to numerous superimpositions. Figure 5 gives an overview of all the pathways leading to the tri-adducts and tetra-adducts. It shows that the “unoriented” and major di-adducts **4** and **5** are converted into the four tri-adducts, presumably with a preference for the *anti* compounds **7a1** and **7a2**. The syn di-adduct **6b** will lead to the two *syn* tri-adducts **7b** and **7c** while each of the *anti* compounds **6a1** and **6a2** will exclusively give a single *anti* tri-adduct, respectively **7a1** and **7a2**. The conversion of the tri-adducts into tetra-adducts offers more possibilities but all *syn* compound **8a**, may only be accessed from **7b** and **7c**. 

The ratio between di-adducts **4** and **5** depends on the polarization of the unsubstituted double bond and this clearly favors compound **5**. Formation of the di-adducts **6** is doubly handicapped since both the steric hindrance and the polarization of the double bond disfavor the vicinal attack. In principle, since there are two reactions leading to the *syn* adduct, the yield should be twice the yield of the *anti* adducts. The same arguments could in principle be applied to the formation of the higher order adducts assuming that the tri-adduct is formed from the di-adducts and the tetra-adduct from the tri-adducts. While this last proposal is certainly true, it cannot be ruled out that reversibility plays a role in these sequences. This is currently under investigation.

## 4. Biological Evaluation of the Compounds

The mono-adduct **3**, the di-adducts **4** and **5** and the mixture of tri-adducts were assayed for antifungal, anti-inflammatory, and anti-bacterial activities as well as cytotoxicity. Only compound **3** was found to have an anti-inflammatory activity (ROS) at the 5 μM level. A cytotoxicity assay was conducted against HeLa (cervical cancer), Au 565 (breast cancer) with 3T3 (mouse fibroblast) as reference. Activity was observed in the 10 μM range for the di-adducts and the tri-adduct, unfortunately with no selectivity (Table 2).

## 5. Materials and Methods

### 5.1. General

Reactions were carried out in an open atmosphere, using distilled solvents and oven dried glassware. Acetonitrile was HPLC grade. Silica gel TLC plates (Merck silica gel 60 F254 plates) were used to monitor the reactions. Column chromatography CC) was carried out on silica gel 60 (Merck, mesh size 70–230, Billerica, MA, USA). UV data were measured on a Thermo-scientific model-300 spectrometer. IR data were recorded on a Bruker Vector-22 spectrophotometer on KBr disk. Optical rotations were recorded on JASCO P-2000 spectrometer. CD spectra were recorded on JASCO J-810 spectropolarimeter. ^1^H-NMR spectra were recorded on AV-500 MHz spectrometers in deuterated methanol. *J*-values (coupling constants) were expressed in Hertz (Hz). ^13^C-NMR spectra were recorded on Bruker AVIII-300, and 600 MHz spectrometers in deuterated chloroform or methanol. ESI-MS spectra were recorded on Bruker mass spectrometer Amazon ESI ion trap (Bruker, Billerica, MA, USA). The high-resolution electrospray ionization spectra (HR-ESI) were recorded on a mass spectrometer Bruker Daltonic Maxis II/ESI Q-TOF system with electrospray ionization (ESI) ion source. 

### 5.2. Preparation of 2-(10-Vindolinyl)-benzoquinone ***3***

In a 50 mL round bottom flask containing 14 mL of a (7/1) mixture of acetonitrile and water, 200.5 mg (0.44 mmole) of vindoline and 1 mL TFA were added. The mixture was stirred for five minutes at room temperature, and 188.9 mg (1.74 mmole) p-benzoquinone was added in the reaction mixture, giving a reddish-brown coloration. The mixture was stirred for 24 h at room temperature, and the color became more pronounced. Acetonitrile was evaporated under reduced pressure and a solution of NaHCO_3_ was added until pH became basic. Then 20 mL CH_2_Cl_2_ was added, and the reaction mixture was transferred to a separating funnel. After shaking, the organic phase was separated. The process was repeated three times. The organic phases were combined, dried over Na_2_SO_4_, filtered and evaporated to give a blue residue, which was purified by flash chromatography with CH_2_Cl_2_ containing 1% MeOH as eluent. Compound **3** (235 mg, 95% yield) was obtained as a blue amorphous solid. G)

*2-(10-Vindolinyl)-benzoquinone***3**, blue amorphous solid, melting point = 250 °C; *R*_f_ = 0.53 (1%/99% MeOH/CH_2_Cl_2_) 0.683;$\;{\left[\alpha \right]}_{D}^{20}$ = +12 (*c* 0.0008, MeOH); UV (λ_max_ MeOH): 221, 250, 309, 542 nm; IR (KBr): 3451, 2928, 2877, 1742, 1662, 1610, 1502, 1442, 1377, 1246, 1092, 1047 cm^−1^; ^1^H NMR (500 MHz, CD_3_OD): δ = 7.04 (1H, s, H-9), 6.86 (1H, d, *J* = 8.0 Hz, H-6′), 6.82 (1H, dd, *J* = 8.0, 2.0 Hz, H-5′), 6.75 (1H, d, *J =* 2.0 Hz, H-3′), 6.34 (1H, s, H-12), 5.91 (1H, ddd, *J =* 10.2, 4.9, 1.4 Hz, H-14), 5.38 (1H, s, H-17), 5.24 (1H, br d, *J* = 10.2 Hz, H-15), 3.81 (6H, s, C-11 OCH_3_ and COOCH_3_), 3.76 (1H, s, H-2), 3.51 (1H, dd, *J* = 16.2, 4.9 Hz, H-3*β*), 3.42 (1H, td, *J* = 9.5, 4.6 Hz, H-5*β*), 2.93 (1H, br d, *J =* 16.2 Hz, H-3*α*), 2.86 (1H, s, H-21), 2.79 (3H, s, N-CH_3_), 2.66 (1H, dt, *J* = 9.5, 7.6 Hz, H-5α), 2.33 (2H, m, 2H-6), 2.04 (3H, s, C-17 OCOCH_3_), 1.64 (1H, dq, *J* = 14.5, 7.2 Hz, H-19a), 1.22 (1H, dq, *J* = 14.5, 7.2 Hz, H-19b), 0.63 (3H, t, *J* = 7.2 Hz, H-18); ^13^C NMR (150 MHz, CDCl_3_): δ = 188.0 (C-1′), 186.5 (C-4′), 171.8 (COOCH_3_), 170.7 (OCOCH_3_), 159.5 (C-11), 155.1 (C-13), 145.1 (C-2′) 137.0 (C-6′) 136.1 (C-5′) 132.8 (C-3′), 130.3 (C-15), 125.0 (C-9), 124.4 (C-8), 124.2 (C-14), 112.6 (C-10), 92.9 (C-12), 83.3 (C-2), 79.4 (C-16), 76.2 (C-17), 66.8 (C-21), 55.8 (C-11 OCH_3_), 52.7 (C-7), 52.3 (COOCH_3_), 51.6 (C-5), 50.9 (C-3), 43.9 (C-6), 42.8 (C-20), 37.6 (N-CH_3_), 30.9 (C 19), 21.6 (OCOCH_3_), 7.6 (C-18); HRESI-MS calc. for: C_31_H_35_N_2_O_8_ [M + H]^+^ *m*/*z* = 563.2388, measured 563.2393. 

### 5.3. Preparation of 2,5-di-(10-Vindolinyl)-benzoquinone ***4***, of 2,6-di-(10-Vindolinyl)-benzoquinone ***5*** and of 2,3-di-(10-Vindolinyl)-benzoquinone ***6***

In a 50 mL round bottom flask containing 4 mL distilled water, 1115 mg (6.56 mmoles) of CuCl_2_·2H_2_O was dissolved, followed by 18.9 mg (0.166 mmole) *p*-benzoquinone, and 200 mg (0.43 mmole) vindoline. After 5 min, 1 mL TFA and 2 mL acetonitrile was added and the mixture was stirred for 8 h at room temperature. The acetonitrile was evaporated under reduced pressure and the pH of the solution made basic with NaHCO_3_. The same operating procedure as above was followed and the mixture was separated by flash chromatography (2% MeOH in CH_2_Cl_2_). Three compounds were thus separated: 2, 5-di-(10-vindolinyl)-1,4 *p*-benzoquinone **4** (16 mg, 9%) 2,6-di-(10-vindolinyl)-1,4 *p*-benzoquinone **5** (134 mg, 75%) and 2,3-di-(10-vindolinyl)-1,4 *p*-benzoquinone **6** (14 mg, 7%).

*2,5-di-(10-vindolinyl)-benzoquinone***4**, pink amorphous solid; melting point 266 °C; *R*_f_ = 0.47 (2%/98% MeOH/CH_2_Cl_2_); ${\left[\alpha \right]}_{D}^{20}\;$= +7.3 (*c* 0.00045, MeOH); UV (λ_max_ MeOH): 223, 255, 312, 524; IR (KBr): 3460, 2926, 2878, 2852, 1743, 1651, 1614, 1505, 1461, 1433, 1374, 1335, 1244, 1158, 1092, 1040, 1047, 948, 892, 816, 747 cm^−1^; ^1^H-NMR (500 MHz, CD_3_OD): δ 7.04 (2H, s, H-9), 6.76 (2H, s, H-3′/H-6′), 6.35 (2H, s, H-12), 5.90 (2H, ddd, *J =* 10.2, 6.7 Hz, 4.6 Hz, H-14), 5.38 (2H, s, H-17), 5.24 (2H, br d, *J* = 10.2 Hz, H-15), 3.83 (6H, s, C-11 OCH_3_), 3.81 (6H, s, COOCH_3_), 3.75 (2H, s, H-2), 3.48 (2H, dd, *J =* 16.4 Hz, 4.6 Hz, H-3*β*), 3.42 (2H, dt, *J =* 8.9, 4.3 Hz, H-5*β*), 2.94 (2H, br. d, *J =* 16.4 Hz, H-3α), 2.89 (2H, s, H-21), 2.79 (6H, s, NCH_3_), 2.66 (2H, m, H-5α), 2.30 (4H, m, H-6), 2.04 (6H, s, OCOCH_3_), 1.63 (2H, dq, *J* = 14.5, 7.2 Hz, H-19a), 1.22 (2H, dq, *J* = 14.5, 7.2 Hz, H-19b), 0.64 (6H, t, *J* = 7.2 Hz, CH_3_-18); ^13^C-NMR (75 MHz, CD_3_OD): *δ* 188.3 (C-1′/C-4′), 173.4 (COOCH_3_), 172.5 (OCOCH_3_), 161.1 (C-11), 156.5 (C-13), 146.7 (C-2′/C-5′), 134.3 (C-3′/C-6′) 131.3 (C-15), 126.6 (C-9), 125.9 (C-8), 125.8(C-14), 114.6 (C-10), 94.4 (C-12), 84.5 (C-2), 80.9 (C-16), 77.6 (C-17), 67.6 (C-21), 56.4 (C-11 OCH_3_), 54.1 (C-7), 52.9 (COOCH_3_), 52.9 (C-5), 52.4 (C-3), 44.6 (C-6), 44.32 (C-20), 38.3 (N-CH_3_), 32.1 (C-19), 20.8 (OCOCH_3_), 8.1 (C-18); HRESI-MS calc. for: C_56_H_65_N_4_O_14_ [M + H]^+^ *m*/*z* 1017.4492, measured 1017.4496. 

*2,6-di-(10-vindolinyl)-benzoquinone***5**, blue amorphous solid; melting point 266 °C; *R*_f_ = 0.43 (2% MeOH/CH_2_Cl_2_); ${\left[\alpha \right]}_{D}^{20}\;$= +37 (c 0.0006, MeOH); UV (λ_max_ MeOH): 222, 275, 310, 550; IR (KBr): 3458.4, 2925, 2877, 2852, 1743, 1641, 1613, 1504, 1462, 1433, 1448, 1430, 1374, 1246, 1166, 1114, 1077, 1040, 950, 892, 816, 785, 741, 644, 615, 584, 545, 481, 422 cm^−**1**^; ^1^H-NMR (500 MHz, CD_3_OD) *δ* 7.02 (2H, s, H-9), 6.75 (2H, H-3′/H-5′), 6.34 (2H, s, H-12), 5.89 (2H, ddd, *J* = 10.2, 5.0, 3.5 Hz, H-14), 5.37 (2H, s, H-17), 5.21 (2H, br d, *J* = 10.2 Hz, H-15), 3.86 (6H, s, C-11 OCH_3_), 3.81 (6H, s, COOCH_3_), 3.76 (2H, s, H-2), 3.50 (2H, dd, *J =* 16.3, 5.0 Hz, H*β*-3), 3.42 (2H, td, *J =* 9.0, 4.3 Hz, H*β*-5), 2.90 (2H, br d, *J* = 16.3 Hz, H*α*-3), 2.83 (2H, s, H-21), 2.79 (6H, s, N-CH_3_), 2.63 (2H, m, H*α*-5), 2.34 (4H, m, H-6), 2.04 (6H, s, OCOCH_3_), 1.64 (2H, dq, *J* = 14.5, 7.2 Hz, Ha-19),1.20 (2H, dq, *J* = 14.5, 7.2 Hz, Hb-19), 0.59 (6H, t, *J* = 7.2 Hz, H-18); ^13^C-NMR (600 MHz, CD_3_OD): *δ* 189.6 (C-1′), 187.8 (C-4′), 173.4 (COOCH_3_), 172.5 (OCOCH_3_), 161.2 (C-11), 156.7 (C-13), 148.7 (C-2′/C-6′), 132.4 (C-3′/C-5′), 131.2 (C-15), 126.5 (C-9), 125.9 (C-8), 125.8 (C-14), 115.5 (C-10), 94.4 (C-12), 84.4 (C-2), 80.8 (C-16), 77.5 (C-17), 67.9 (C-21), 56.6 (C-11 OCH_3_), 54.0 (C-7), 52.9 (COOCH_3_), 52.6 (C-5), 52.0 (C-3), 44.6 (C-6), 44.3 (C-20), 38.3 (N-CH_3_), 32.1 (C 19), 20.8 (OCOCH_3_), 8.1 (C-18); HRESI-MS calc. for: C_56_H_65_N_4_O_14_ [M + H]^+^ *m*/*z* 1017.4492, measured 1017.4495.

*2,3-di-(10-vindolinyl)-benzoquinone***6**, brown amorphous solid; melting point 267 °C *R*_f_ = 0.36 (2% MeOH/CH_2_Cl_2_); ${\left[\alpha \right]}_{D}^{20}$ = −10.8 (c 0.001, MeOH). UV (λ_max_ MeOH): 300, 534 nm; IR (KBr): cm^−**1**^. *Anti* isomer **6a**: ^1^H NMR (500 MHz, CD_3_OD) *δ* 6.89 (2H, s, H-5′/H-6′), 6.31 (2H, s, H-9), 6.28 (2H, s, H-12), 5.85 (2H, ddd, *J* = 10.2, 5.0, 3.5 Hz, H-14), 5.35 (2H, s, H-17), 5.18 (2H, br d, *J* = 10.2 Hz, H-15), 3.80 (6H, s, C-11 OCH_3_), 3.79 (6H, s, COOCH_3_), 3.58 (2H, s, H-2), 3.45 (2H, m, H-3*β*), 2.82 (2H, br d, *J* = 16.3 Hz, H-3*α*), 2.67 (6H, s, N-CH_3_), 2.54 (2H, s, H-21), 2.02 (6H, s, OCOCH_3_), 1.62 (2H, m, H-19a),1.15 (2H, m, H-19b), 0.47 (6H, t, *J* = 7.2 Hz, H-18); ^13^C NMR (150 MHz, CD_3_OD): δ 187.5 (C-1′, C-4′), 173.4 (CO_2_CH_3_), 172.4 (OCOCH_3_), 160.6 (C-11), 155.5 (C-13), 145.2 (C-2′/C-3′), 137.4 (C-5′/C-6′), 131.1 (C-15), 125.8 (C-9), 125.7 (C-8), 125.6 (C-14), 116.0 (C-10), 94.1 (C-12), 84.6 (C-2), 80.8 (C-16), 77.5 (C-17), 68.0 (C-21), 56.6 (C-11 OCH_3_), 53.6 (C-7), 52.9 (COOCH_3_), 52.7 (C-5), 51.9 (C-3), 44.4 (C-6), 44.2 (C-20), 38.7 (N-CH_3_), 32.1 (C 19), 20.8 (OCOCH_3_), 8.2 (C-18). *Syn* isomer **6b:** ^1^H NMR (500 MHz, CD_3_OD), most characteristic peaks *δ* 6.88 (1H, s, H-5′ or H-6′), 6.87 (1H, s, H-3′ or H-5′), 6.84 (1H, s, H-9), 6.67 (1H, s, H-9), 6.24 (1H, s, H-12), 6.13 (1H, s, H-12), 5.43 (1H, s, H-17), 5.38 (1H, s, H-17), 3.67 (3H, s, C-11 OCH_3_), 3.79 (6H, s, COOCH_3_), 3.55 (3H, s, C-11 OCH_3_), 3.64 (1H, s, H-2), 3.52 (1H, s, H-2), 2.69 (3H, s, N-CH_3_), 2.65 (3H, s, N-CH_3_), 2.77 (1H, s, H-21), 2.55 (1H, s, H-21), 2.03 (3H, s, OCOCH_3_), 2.02 (3H, s, OCOCH_3_), 0.64 (3H, t, *J* = 7.2 Hz, CH_3_-18), 0.31 (3H, t, *J* = 7.2 Hz, CH_3_-18); ^13^C NMR (150 MHz, CD_3_OD) most characteristic peaks distinct from those of **6a**: 126.4 (C-9), 126.1 (C-9), 125.7 (C-8), 125.6 (C-14), 116.0 (C-10), 94.1 (C-12), 84.6 (C-2), 80.8 (C-16), 77.5 (C-17), 68.0 (C-21), 56.6 (C-11 OCH_3_), 53.6 (C-7), 52.9 (COOCH_3_), 52.7 (C-5), 51.9 (C-3), 44.4 (C-6), 44.2 (C-20), 38.7 (N-CH_3_), 32.1 (C 19), 20.8 (OCOCH_3_), 8.2 (C-18).

### 5.4. Preparation of 2,3,5-tris-(10-Vindolinyl)-benzoquinone ***7*** and of 2,3,5,6-tetra-(10-Vindolinyl)-benzoquinone ***8***

In a 50 mL round bottom flask containing 4 mL distilled water, dissolve 560 mg (3.3 mmoles) of CuCl_2_·2H_2_O, followed by 9.5 mg (0.087 mmole) p-benzoquinone, and 150.8 mg (0.33 mmole) vindoline. After 5 min, 1 mL TFA and 2 mL acetonitrile were added and the mixture was stirred for 48 h at room temperature. The acetonitrile was evaporated under reduced pressure and the pH of the solution was made basic with NaHCO_3_. The same operating procedure as above was followed and the mixture was separated by flash chromatography (3% MeOH in CH_2_Cl_2_). Three compounds were thus separated: 2, 6-di-(10-vindolinyl)-1,4 *p*-benzoquinone **5** (6 mg, 5%), 2,3,5-tri-(10-vindolinyl)-1,4 *p*-benzoquinone **7**, (60 mg, 37%), 2,3,5, 6-tetra-(10-vindolinyl)-1,4 *p*-benzoquinone **8** (26 mg, 15%)

*2,3,5-tris-(10-vindolinyl)-benzoquinone* **7**, pink amorphous solid; melting point 275 °C; *R*_f_ = 0.54 (3%/97% MeOH/CH_2_Cl_2_); ${\left[\alpha \right]}_{D}^{25}\;$= +241 (c 0.0023, MeOH); UV (λ_max_ MeOH): 229, 261, 310, 519, 529; IR (KBr): 3431, 2946, 1739, 1613, 1500, 1448, 1374, 1242, 1107, 1035, 889, 820, 743, 628, 481, 584, cm^−1^; ^1^H-NMR (500 MHz, CD_3_OD) major isomer δ *=* 7.07(3H, s, H-9), 6.81 (3H, s, H-6′), 6.36 (3H, s, H-12), 6.35 (3H, s, H-9), 6.32 (3H, s, H-9), 6.30 (3H, s, H-12), 6.29 (3H, s, H-12), 6.23, 6.18 (3H, s, H-12), 5.90–5.82 (3H, m, H-14), 5.38, 5.36, 5.35 (3H, 3s, H-17), 5.25–5.15 (3H, m H-15), 3.85, 3.82, 3.81 (9H, 3s, C(11)OCH_3_), 3.80, 3.79 (9H, s, C(16)COOCH_3_), 3.77, 3.59, 3.58 (3H, 3s, H-2), 2.88, 2.57, 2.50 (3H, 3s, H-21), 2.79, 2.68, 2.67 (9H, s, NCH_3_), 2.04, 2.02, 2.01 (9H, 3s, C(17)OCOCH_3_), 0.64, 0.48, 0.44, (9H, 3t, *J* = 7.2 Hz, CH_3_-18); ^13^C-NMR (150 MHz, CD_3_OD) major isomer δ *=* 187.9 (C-1′), 187.0 (C-4′), 173.4, (3 C-16-COOCH_3_), 172.43, 172.41 (3 C-17-OCOCH_3_), 161.3, 160.7, 160.5 (3 C-11), 156.6, 155.4, 155.3 (3 C-13), 147.9 (C-2′ or C-5′), 146.2 (C-3′) 144.3 (C-2′ or C-5′), 133.2 (C-6′), 131.2, 131.1, 131.0 (3 C-15), 126.5, 126.0, 125.9 (3 C-9), 125.8, 125.6 (3 C-14),125.9, 125.8 (3 C-8), 117.0, 116.2, 115.4 (3 C-10), 94.5, 94.2, 94.1, (3 C-12), 84.61, 84.60, 84.5, (3 C-2), 80.86, 80.85, 80.78 (C-16), 77.56, 77.53, 77.51 (3 C-17), 68.1, 67.9, 67.7 (3 C-21), 56.8, 56.65, 56.64, (3 C-11OCH_3_), 54.1, 53.65, 53.55 (3 C-7), 52.9 (3 C-16-COOCH_3_), 52.8, 52.6, 52.5, (3 C-5), 52.0, 51.9, 51.8 (3 C-3), 45.5, 45.4, 44.7 (3 C-6), 44.7, 44.55, 4.45 (3 C-20), 38.8, 38.4 (3 C-1), 32.2, 32.1 (3 C 19), 20.8, 20.7 (3 C-17-OCOCH_3_), 8.23, 8.1 (3 C-18); HRESI-MS calc. for: C_81_H_95_N_6_O_20_ [M + H]^+^ *m*/*z* 1471.6596, measured 1471.6633.

*2,3,5,6-tetra-(10-vindolinyl)-benzoquinone***8**, pink amorphous solid; melting point 280 °C; *R*_f_ = 0.47 (3%/97% MeOH/CH_2_Cl_2_); ${\left[\alpha \right]}_{D}^{25.94}\;$= -9.5 (c 0.0023, MeOH); UV (λ_max_ MeOH): 229, 260, 310, 515; IR (KBr): 3608, 2957, 2879, 1738, 1614, 1500, 1449, 1372, 1244, 1040, 824, 746, 628 cm^−**1**^; ^1^H-NMR (500 MHz, CD_3_OD) δ *=* 7.01, 6.94, 6.92, 6.90, 6.82, 6.80, 6.87, 6.75, 6.68, 6.45, 6.40, 6.36, 6.34, 6.31 (4H, s, H-9), 6.28, 6.26, 6.22, 6.20, 6.17, 6.13, 6.12, 6.10 (4H, s, H-12), 5.85 (4H, m, H-14), 5.46, 5.42, 5.37, 5.36, 5.34 (3H, s, H-17), 5.28, 5.23, 5.12, 516 (4H, m, H-15), 3.87, 3.85, 3.83, 3.79, 3.76, 3.75 (24 H, s, C(11)OCH_3_, C(16)COOCH_3_), 3.76, 3.60 (4H, m, H-2), 3.50 (4H, *m*, H*β*-3), 3.47–3.45 (4H, *m*, H*β*-5), 2.89–2.73 (4H, m, H*α*-3), 2.64, 2.59, 2.56,2.53 (4H, s, H-21), 2.67 (12H, s, H-1), 2.45–2.2.40, (4H, *m*, H*α*-5), 2.30–2.06 (8H, m, H-6), 2.061 (12H, s, C(17)OCOCH_3_), 1.58–1.54 (4H, m, Ha-19),1.15–1.10 (3H, *m*, Hb-19), 0.57, 0.47, 0.41, 0.39, 0.30 (4H, overlap t, H-18); HRESI-MS calc. for: C_106_H_125_N_8_O_26_ [M + H]^+^ *m*/*z* 1925.8700, measured 1925.8702.

### 5.5. Computational Details

Considering methanol as an implicit solvent (CPCM [16,17]), the systems under study were optimized at the DFT (M06-2X [18]) level of theory and using the 6-311G** basis set within the restricted formalism. The program Gaussian-16 [19] was employed. For the transition states (TS), each geometry was characterized by the presence of a single imaginary vibrational frequency and IRC calculations have been performed to check that the TS connects the two desired minima. Reaction and activation free energies were obtained using the KiSThelP program [20]. 

## Data Availability

Full NMR data are available upon request from S.A. or G.M.

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
