# Peer review of "Addition of Vindoline to p-Benzoquinone: Regiochemistry, Stereochemistry and Symmetry Considerations"

_molecules, 2021, doi:10.3390/molecules26216395_

Round 1

Reviewer 1 Report

The authors did not cite important results in the topic of the manuscript. I would suggest to cite two relevant papers: 

Zhang et al  (Adv. Synth. Catal.  2006, 348, 229-235.)  prepared aryl-substituted benzoquinones by the addition reactions of aromatic compounds in water solvent using indium(III)triflate catalysis.  

C – H arylation of benzoquinone was performed with electron rich arenes (aromatic tert amines or phenolethers)  using triflic acid catalyst. Jiang et al Eur. J. Org. Chem. 2016, 2284-2289. 

If you compare of the chemical shifts of CH3-18, H-14 and H-21 in compound 3 (monoadduct) with that of compounds 4 and 5 the differences (max 0.06 ppm) are not significant in my opinion. (page 4  line 137)  I would mention the only example that the 13C chemical shifts of the quinone moiety (C1’ and C4’) are not differentiated in compound 4.  

 The presentation of the NMR spectra of compound 6 is not consistent the 13C spectrum of the syn isomer was listed first and after the 1H spectrum  again. 

The separation of compounds 5 and 6 was successful but  the Rf values are identical. Otherwise the purity of the prepared compounds was not determined, I miss the HPLC studies.   

Author Response

The authors did not cite important results in the topic of the manuscript. I would suggest to cite two relevant papers: 

Zhang et al (Adv. Synth. Catal.  2006, 348, 229-235.)  prepared aryl-substituted benzoquinones by the addition reactions of aromatic compounds in water solvent using indium (III)triflate catalysis.  

C – H arylation of benzoquinone was performed with electron rich arenes (aromatic tert amines or phenolethers)  using triflic acid catalyst. Jiang et al Eur. J. Org. Chem. 2016, 2284-2289. 

We have added these two references with some comments. The first one gives mono adduct and di-adduct depending on reaction time and stoichiometry. The di-adduct structure is analogous to the one of our main compound 5 but no further adducts were described. The second article combines oxidation of a phenol to a benzoquinone followed by arylation catalyzed by triflic acid. Only mono adducts were obtained. In these two articles, the authors rely on air and excess benzoquinone to oxidize the intermediate dihydroquinones. The addition of CuII, which we propose here, might improve their yields.

If you compare of the chemical shifts of CH3-18, H-14 and H-21 in compound 3 (monoadduct) with that of compounds 4 and 5 the differences (max 0.06 ppm) are not significant in my opinion. (page 4  line 137)  

This is exactly what we said: “The 1H NMR spectra are not very different either, noticeable and unexplained differences being observed for CH3-18 (0.05 ppm), H-14 (0.03ppm) and H-21 (0.06 ppm)”. We wish to maintain this sentence because the differences exist and might be explained some day using more sophisticated models. If one considers vindoline as a mostly flat molecules, Me-18 and H-21 are located on the same side of this plane and are therefore probes for the characterization of this particular side, which has some importance on stacked adducts such as 6a and 6b.

I would mention the only example that the 13C chemical shifts of the quinone moiety (C1’ and C4’) are not differentiated in compound 4. There is an axis of symmetry which exchanges the two quinone carbonyls in compound 4, therefore their resonances should be the same.

 The presentation of the NMR spectra of compound 6 is not consistent the 13C spectrum of the syn isomer was listed first and after the 1H spectrum  again. 

This is our fault and the redundant data were removed.

The separation of compounds 5 and 6 was successful but  the Rf values are identical. Otherwise the purity of the prepared compounds was not determined, I miss the HPLC studies.

RF of 5 is 0.43; Rf of 6 is 0.36. Thank you for pointing out this mistake. UPLC will be added to SI  

Reviewer 2 Report

In this article, Ali and co-workers investigate the vindoline reaction with benzoquinone as an electrophile. The authors have optimized the reaction of vindoline with benzoquinone by changing the stoichiometric ratios of vindoline and benzoquinone in Table 1. Interestingly the 1:4 ratio of vindoline and benzoquinone gives the highest yield of the mono adduct (compound 3). However, the high stoichiometry of vindoline gives the poly adduct (di, tri tetra) of the product (compounds 4, 5, 7) (Table 1 entries 9–14). The authors used CuCl2 for the formation of high addition and which gives the high addition product (Table 1, entries 9–12). In this manuscript, the authors have distinguished all these adducts using column chromatography techniques and they determine the structure of these adducts using 1D and 2D NMRs and symmetry correlation. In addition, the authors have also demonstrated the biological activity of these molecules and investigated the cytotoxicity of these compounds against HeLa, Au565 cancer, and 3T3 normal cell lines. And the authors observed that compound 3 shows an anti-inflammatory activity, however, compounds 4 and 5 show 10 uM inhibition concentration without any selectivity (Table 2). All of these are well presented in this manuscript and will be considered for publication in Molecules after the revision.

Recommendation on the minor revision:

  • The author mentioned in the manuscript the adducts structures are determined by using HMBC, HSQC, and NOESY in SI. Please assign each peak and clearly show their correlation in the revised manuscript and SI.
  • In SI none of the 1H NMR peaks are integrated, please integrate all the 1H NMR peaks and do the peak labeling for each and 1H and 13C NMR traces presented in SI.

Author Response

In this article, Ali and co-workers investigate the vindoline reaction with benzoquinone as an electrophile. The authors have optimized the reaction of vindoline with benzoquinone by changing the stoichiometric ratios of vindoline and benzoquinone in Table 1. Interestingly the 1:4 ratio of vindoline and benzoquinone gives the highest yield of the mono adduct (compound 3). However, the high stoichiometry of vindoline gives the poly adduct (di, tri tetra) of the product (compounds 4, 5, 7) (Table 1 entries 9–14). The authors used CuCl2 for the formation of high addition and which gives the high addition product (Table 1, entries 9–12). In this manuscript, the authors have distinguished all these adducts using column chromatography techniques and they determine the structure of these adducts using 1D and 2D NMRs and symmetry correlation. In addition, the authors have also demonstrated the biological activity of these molecules and investigated the cytotoxicity of these compounds against HeLa, Au565 cancer, and 3T3 normal cell lines. And the authors observed that compound 3 shows an anti-inflammatory activity, however, compounds 4 and 5 show 10 uM inhibition concentration without any selectivity (Table 2). All of these are well presented in this manuscript and will be considered for publication in Molecules after the revision.

Recommendation on the minor revision:

  • The author mentioned in the manuscript the adducts structures are determined by using HMBC, HSQC, and NOESY in SI. Please assign each peak and clearly show their correlation in the revised manuscript and SI. We are afraid that doing so would considerably and unnecessarily lengthen the size of our manuscript. We propose instead of assigning peaks and showing correlation to use an appendix on the assignment of vindoline in our adducts. The 13C NMR spectrum of vindoline was assigned by Wenkert in 1973 and at > 300 MHz frequency, the 1H NMR can be readily deciphered.
  • In SI none of the 1H NMR peaks are integrated, please integrate all the 1H NMR peaks and do the peak labeling for each and 1H and 13C NMR traces presented in SI. We did that to make our spectra more legible but we are going to add integration on the proton spectra when useful (not for tri- and tetra-adducts) and peak picking. If the Journal prefers to build a repository with crude data, we think that will be by far the best solution.

Reviewer 3 Report

This is an interesting study of various synthetic vindoline adducts involving some truly intricate stereochemistry. The paper is well written, the (rather complex) stereochemical considerations are sound, the experimental material (especially the NMR data) and the conclusions drawn from them are correct. I only have some minor corrections that I would recommend for consideration by the authors in order to further improve this manuscript.

line 24: for the sake of precision, replace “on the NMR time scale” with “on the NMR chemical shift timescale”.

lines 44-47: this is somewhat misleading, since VLB and VCR are also present naturally in Catharanthus roseus, and VLB can be produced on an industrial scale by direct extraction and subsequent purification. VCR is also often produced via the synthetic oxidation of the so obtained VLB.

lines 154/155: “are one and the same compound” may be more technically phrased as “are homomers”.

line 170: replace “where the H-2 are either” with “where the H-2 atoms are either”.

line 245: replace “are due to an exchange process” with “are due to a slow exchange process”.

line 269: repace “there” with “they”

line 270: “case, either” → “case either”

Author Response

This is an interesting study of various synthetic vindoline adducts involving some truly intricate stereochemistry. The paper is well written, the (rather complex) stereochemical considerations are sound, the experimental material (especially the NMR data) and the conclusions drawn from them are correct. I only have some minor corrections that I would recommend for consideration by the authors in order to further improve this manuscript.

line 24: for the sake of precision, replace “on the NMR time scale” with “on the NMR chemical shift timescale”. OK

lines 44-47: this is somewhat misleading, since VLB and VCR are also present naturally in Catharanthus roseus, and VLB can be produced on an industrial scale by direct extraction and subsequent purification. VCR is also often produced via the synthetic oxidation of the so obtained VLB. We catch the point and propose to write: Vindoline and catharanthine are extracted on an industrial scale in a process aiming at preparing the pharmacologically important navelbine. VLB, VCR are simultaneously obtained and since the amount of vindoline exceeds the amount of catharanthine, it is of interest to find new uses for vindoline.

lines 154/155: “are one and the same compound” may be more technically phrased as “are homomers”. This is not a word we are used to, but let us try.

line 170: replace “where the H-2 are either” with “where the H-2 atoms are either”. OK done

line 245: replace “are due to an exchange process” with “are due to a slow exchange process”. OK done

line 269: repace “there” with “they” OK done

line 270: “case, either” → “case either” OK done

Round 2

Reviewer 2 Report

The authors have addressed most of the concerns raised previously. And it is recommended for publication in Molecules.